



# Enhancing forest air sampling using a novel reusable ozone filter design

Robby Rynek[1], Thomas Mayer[1], Helko Borsdorf[1]

[1] Department of Monitoring and Exploration Technologies, Helmholtz Centre for Environmental Research GmbH – UFZ,
04318 Leipzig

*Correspondence to*: Robby Rynek (robby.rynek@ufz.de)

**Abstract.** Biogenic volatile organic compounds (BVOCs), such as monoterpenes, play essential roles in ecological and atmospheric processes, influencing air quality, climate and interspecies interactions. For accurate identification and quantification of these reactive compounds in the environment, active sampling on sorbent tubes followed by thermodesorption gas chromatography-mass spectrometry is commonly used. However, ozone present in the sampled air can degrade both the analytes and the sorbent material during the sampling process, leading to underestimation of target substances and overestimation of their degradation products. This study evaluates a novel reusable ozone filter designed for direct attachment to sorbent tubes and compatibility with multi-tube samplers. The filter utilizes potassium iodide (KI) or sodium thiosulfate ($Na_2S_2O_3$) deposited on reusable glass filters and copper wool to improve the accuracy of BVOC measurements. Both types of ozone scrubbers were tested under varying ozone concentrations up to 50 ppb and relative humidity levels up to 90 %, utilizing a straightforward load-and-flush method as well as a permeation approach that simulates field sampling conditions. Furthermore, both methods were compared regarding their suitability for the systematic evaluation of ozone filters.

Results indicate, that both KI and $Na_2S_2O_3$ effectively remove ozone, with KI showing a slightly higher performance and lower dependence on relative humidity, maintaining over 90 % removal efficiency even after 10 days of ambient air exposure. Recovery rates for four structurally different monoterpenes (α-Pinene, Myrcene, Limonene, Linalool) showed no significant differences between filtered and unfiltered samples at baseline ozone concentrations, demonstrating that the ozone filters did not negatively impact analyte recovery. When no filter was used, recovery rates for Myrcene, Limonene, and Linalool declined with increasing ozone concentration, while showing a method-dependent positive influence of increasing relative humidity. Both scrubber materials maintained high and comparable recovery rates across all tested conditions, except at very low relative humidity, thereby enhancing measurement accuracy and comparability under diverse environmental scenarios. Field tests confirmed the effectiveness of KI-loaded scrubbers in enhancing monoterpene detection in forest air while safeguarding the sorbent material. These results, combined with the easy reusability of the glass filters and the absence of additional equipment or power requirements, highlight that this scrubber design proves to be an optimal choice for the long-term environmental monitoring of volatile organic compounds.



## 1 Introduction

Biogenic volatile organic compounds (BVOCs) are a diverse group of organic substances emitted into the atmosphere by a wide range of organisms, with plants representing a significant source (Guenther et al., 1995, 2012; Sindelarova et al., 2014). These compounds, which include isoprene, monoterpenes and sesquiterpenes, play a crucial role in ecological and atmospheric processes. On the one hand, BVOCs significantly influence the chemical composition of the atmosphere by contributing to the formation of ozone and secondary organic aerosols (SOAs), affecting both climate and air quality (Hallquist et al., 2009). On the other hand, they are involved in communication and interaction mechanisms between organisms within a species or across species boundaries and in plants' defence mechanisms against biotic and abiotic stress, underlining their ecological importance (Laothawornkitkul et al., 2009).

In order to investigate and understand both the atmospheric and ecological processes, it is essential to conduct a comprehensive analysis of concentrations and the composition of these compounds in the environment. A common method for obtaining both qualitative and quantitative information on BVOCs in air samples combines active or passive sampling with thermal desorption-gas chromatography-mass spectrometry (TD-GC-MS). In this approach, BVOCs present in the air are enriched on a suitable sorbent material over a sampling period ranging from minutes to hours. Subsequently, the enriched compounds are thermally desorbed and separated based on their respective physico-chemical properties, followed by identification and quantification based on their compound-specific mass-to-charge ratios. By enriching large volumes of air, this method offers high sensitivity down to concentrations in the µg m⁻³ range and high specificity, enabling the detection and differentiation of compounds, including isomers (Borsdorf et al., 2023; Hellén et al., 2024). Although this approach has certain benefits, it is not without its shortcomings. Aside from the time-consuming processes of sample collection, transportation to the laboratory, analysis and data evaluation, the accuracy of the measurements can be affected by environmental conditions during sampling, such as humidity and ozone.

While carbon-based sorbents are negatively affected by humidity levels above 40% relative humidity (rH), hydrophobic polymer-based sorbents like Tenax TA work well over a wide relative humidity range (Maceira et al., 2017). Additionally, ozone can degrade the sorbent material because of its high reactivity, leading to the formation of artefacts and breakdown products (Hwan Lee et al., 2006), which may complicate the identification and quantification of target substances by co-elution and enhanced background noise. Furthermore, the adsorbents efficiency may be negatively affected, leading to underestimation of the BVOC concentrations and lower recovery rates. Besides the reaction with the sorbent material, adsorbed BVOCs with one or more C–C double bond may be decomposed by ozonation, leading to loss of analytes and possible detection of transformation products (Calogirou et al., 1996). Hence, the removal of ozone before it gets in contact with the sorbent or adsorbed BVOCs is a crucial step to enhance the measurements accuracy. To date, various ozone scrubber materials and methods have been used, such as heated stainless-steel tubes (Hellén et al., 2012), NO titration (Pollmann et al., 2005), manganese oxide (Fick et al., 2001), sodium thiosulfate (Ernle et al., 2023) or potassium iodide/copper (Borsdorf et al., 2023; Hellén et al., 2024). Although the different ozone scrubbers are able to effectively remove ozone, they have their specific



disadvantages, such as loss of some analytes (NO titration, manganese oxide), safety concerns (NO titration), the need for additional devices and power supply (NO titration, heated stainless-steel tubes), limited compatibility with multi-tube samplers

and the necessity of regular checks and replacement during measurement campaigns (Hellén et al., 2024; Pollmann et al., 2005).

Various approaches have been employed to assess the influence of ozone and humidity on analyte recovery or to evaluate the efficiency of ozone filters. Methods that simulate realistic environmental concentrations include the diffusion of gas-phase analyte molecules through a glass tube (Calogirou et al., 1996; Helmig et al., 2003; Pollmann et al., 2005) or the introduction

of diluted gas standard mixtures (Ernle et al., 2023; Mermet et al., 2019), both analogous to the permeation approach used in this study. In contrast, approaches involving the direct addition of analytes or methanolic solutions (Fick et al., 2001; Hellén et al., 2024) may introduce uncertainties due to artificially high analyte concentrations or the presence of solvent molecules. Furthermore, loading standard mixtures onto sorbent tubes and subsequently flushing them with ozone-containing humid air enables the investigation of post-adsorption analyte degradation during sampling. To the best of our knowledge, no study has

yet been conducted that has systematically evaluated the influence of ozone filters both on the entire sampling process and post-adsorption degradation processes depending on ozone concentration and relative humidity.

For the application in environmental monitoring using multi-tube samplers, ozone scrubbers must be compact, cost effective, easy to use, stable against various and changing environmental conditions, and highly efficient in preventing analyte degradation while maintaining the integrity of collected samples. In this study, we developed, optimized and validated an

ozone filter tailored for such sampling setups, that avoids the disadvantages of previous ozone filter approaches. Our ozone scrubber can be directly placed before the inlet of a common sorbent tube and equipped with different ozone-depleting reagents (KI or $Na_2S_2O_3$). Our primary objective was to assess the ozone removal efficiency of these scrubbers and their suitability for long-term environmental sampling of terpenes in forest air. To achieve this, we compared the breakthrough of ozone under realistic environmental conditions for typical sampling durations and assessed the long-term stability of the filters.

Furthermore, we systematically evaluated the influence of both scrubber materials on analyte recovery under different environmental conditions using two complementary methods: a load-and-flush approach, where analytes are preloaded onto sorbent tubes and subsequently exposed to ozone-containing humid air, and a permeation approach, which more closely simulates a real-world sampling scenario by enriching analytes from an ozone- and humidity-controlled air stream. Additionally, we aim to determine the most suitable validation approach for ozone scrubber to provide a reliable assessment

of the ozone scrubber performance in field applications. Finally, the proposed ozone scrubber setup was tested under real environmental conditions.





## 2 Materials and Methods

**2.1 Chemicals and Materials**

Methanol (SupraSolv for GC-MS, Merck KGaA, Germany) was used for preparing standards and cleaning of ozone scrubbers. Ultrapure, particle free Milli-Q water (Milli-Q Direct 8, Merck KGaA, Germany) was used to clean the sintered glass filters. Ozone scrubbers were loaded with Potassium iodide (KI, >99.0 %, Th. Geyer GmbH & Co. KG, Germany) or Sodium thiosulfate ($Na_2S_2O_3$, >99.5 %, pentahydrate, Riedel-deHaën, Germany). Zero air was generated using a laboratory compressor

(SICOLAB 062, Dürr Technik GmbH & Co. KG, Germany). The relative humidity of air streams was adjusted by passing through ultrapure water (LiChrosolv for chromatography, Merck KGaA, Germany). Copper wool (~99 %, Carl Roth GmbH & Co. KG, Germany) was placed between the tube caps and ozone scrubbers to prevent reactive iodine produced during ozone removal with KI from entering the sorption tubes. "Bio-monitoring" sorbent tubes (Markes International Ltd., UK) were used for all experiments and pre-conditioned according to the manufacturer's instructions using a TC-20 tube conditioner (Markes

International Ltd., UK). Hydrogen carrier gas was produced in-lab using a HyGen200 hydrogen generator (Claind srl, Italy). Myrcen, Limonene, α-Pinene (all Sigma Aldrich, USA) and Linalool (Glentham Life Sciences Ltd., UK) were used as reference substances. Terpenes MegaMix Standard #1 (Restek GmbH, Germany) was used as calibration standard and Bromobezene-d5 (DeuChem GmbH, Germany) as internal standard.

**2.2 Ozone scrubber design**

The small ozone scrubber has been designed to be compatible with commonly used adsorbtion tubes and diffusion-locking caps (Figure 1), making it compatible with multitube samplers. A glass filter, loaded with the ozone depleting reagent, is encased in a stainless-steel housing with a screw cap, thereby facilitating rapid and straightforward replacement of the filter. Two O-rings, located on top of the glass filter and between the housing and the diffusion-locking cap, prevent the sampled air from bypassing the filter. Furhtermore, copper wool, positioned beneath the glass filter, inhibits the entry of reactive iodine

species into the adsorption tube when KI is used as the scrubber material and serves as an additional reaction site for ozone depletion. By easily replacing the glass filter and the copper wool, the ozone scrubber can be prepared and reused for the next sampling within a few minutes. The reusable glass filters can easily be cleaned by rinsing with water and methanol.





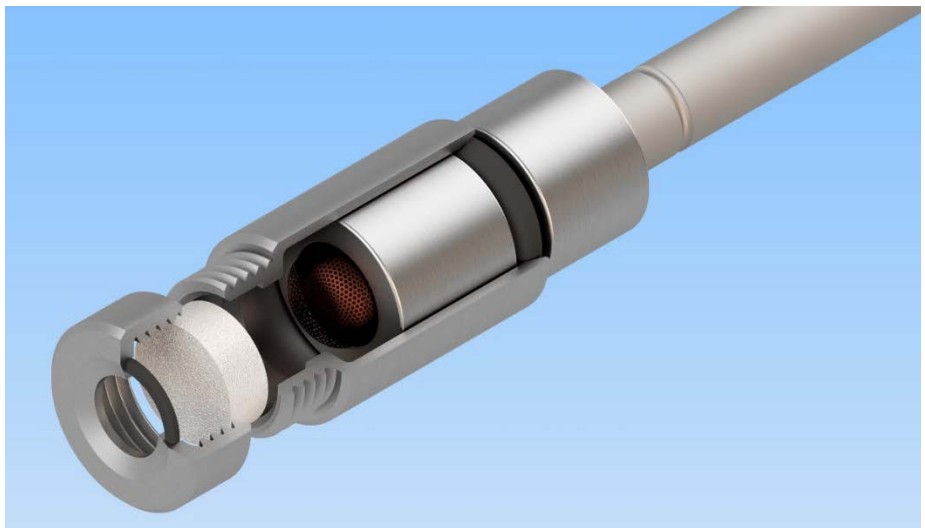

**Figure 1. Schematic layout of the ozone scrubber. The scrubber housing was designed to fit on a commonly used diffusion-locking**
**cap. Sample air flows from the left to the right, passing both a loaded glass filter as ozone scrubber and copper wool. A screw cap**
**facilitates the quick and easy exchange of the filter. Two O-rings, located between the screw cap and the filter and between the**
**housing and the diffusion-locking cap, prevent sampled air from bypassing the filter.**

### 2.3 Preparation of ozone filters

Ozone scrubbers were prepared by loading cleaned and dried glass filters (VitraPOR, diameter 9 mm, height 3.5 mm, pore

diameter 100 - 160 µm, ROBU Glasfilter-Geräte GmbH, Germany) with freshly prepared solutions of KI (10% w/w) or
$Na_2S_2O_3$ (10% w/w). Therefore, the glass filters were placed in a special designed 3D-printed cleaning device, made of Cyclic
olefin copolymer (IWK Institut für Werkstofftechnik und Kunststoffverarbeitung, Switzerland), and rinsed with the solution
until completely wetted. Afterwards, the filters were placed on filter paper to remove excess solution and dried over night at
room temperature. After use, the filters were soaked in ultrapure Milli-Q water, flushed with 100 mL of Milli-Q water each

using the 3D-printed cleaning device, rinsed with methanol in an ultrasonic bath and dried overnight.

### 2.4 Characterization of ozone filters

### 2.4.1 Deposition of KI and $Na_2S_2O_3$ on glass filters

To quantify the amount of the ozone-depleting reagent deposited on the filters, glass filters were weighed before and after
preparation using an ultra-micro balance (MT5, Mettler-Toledo GmbH, Germany). The masses and stochiometric amounts of

KI and $Na_2S_2O_3$ deposited on the filters were calculated and assessed using statistical tests. Additionally, the theoretical ozone
removal potential of both scrubber types was compared.



### 2.4.2 Ozone breakthrough

Ozone was produced photo-chemically using a Stable Ozone Generator (Analytik Jena GmbH & Co. KG, Germany) and continuously quantified with an APOA-370 air pollution monitor (HORIBA Europe GmbH, Germany). Relative humidity was
monitored using a Testo 435-4 multifunction indoor air quality meter (Testo SE & Co. KG, Germany). To evaluate the breakthrough time of both scrubber materials under different conditions, 10 prepared glass filters and copper wool were placed in a 3D-printed filter holder which was designed to meet the flow requirements of the ozone monitor and to ensure a homogeneous flow through all filters (Figure S1). Relative humidity and ozone concentration were adjusted by combining flows of dry, humid and ozone-containing air using mass flow controllers. The filter holder was placed between the outlet of
the combined air flow and the ozone analyser. Breakthrough tests for both scrubber materials and copper wool at a constant input ozone concentration of 50 ± 3 ppb and varying relative humidity values between 1 % and 90 % at 20 – 23 °C were performed over a time of 10 h (Table S1).

### 2.4.3 Long-term stability test

In order to assess the long-term stability and ozone reduction capability of both scrubber materials during a field sampling
campaign, 20 freshly prepared ozone filters of each scrubber material were placed horizontally in a 3D-printed filter holder plate. The plate was then positioned horizontally in inside a sampler box and exposed to a continuous flow of ambient air across the filters surface, simulating the exposure during a real sampling scenario. Ozone concentration, temperature, relative humidity and barometric pressure were monitored during the exposure experiment (Table S2). After 5 and 10 days, respectively, 10 filters per scrubber material were transferred to the laboratory and the breakthrough experiment according to
Sect. 2.4.2 was performed with an input ozone concentration of 50 ppb and a relative humidity of 60%. The removal efficiency of the exposed filters was compared to the freshly prepared ozone scrubbers.

### 2.5 Load-and-flush method

Samples were loaded onto the Bio-monitoring sorbent tubes by injecting 20 μL of a standard solution containing α-Pinene, Limonene, Linalool and Myrcene with concentrations of 1.13 ng/μl, 1.10 ng/μl, 0.99 ng/μl and 1.00 ng/μl, respectively, using
a CIS injection system (Gerstel GmbH & Co. KG, Germany) with a start temperature of 120 °C and heating up to 200 °C. A continuous flow of 100 mL/min through the injector and sorbent tube was realized for 5 min using an air sampling pump (AMA Instruments GmbH, Germany). Resulting masses of the analytes per sorbent tube were 22.6 ng α-Pinene, 22.0 ng Limonene, 19.9 ng Linalool and 20.0 ng Myrcene. Triplicates of loaded sorbent tubes were placed in the sample extraction setup (Figure S2) and flushed for 4 h and a flow of 80 mL/min with air at different levels of ozone concentration and relative
humidity (Table S3). Each sample was additionally loaded with 50 ng of bromobenzene-D5 as internal standard. Two additional loaded sorbent tubes per combination of ozone concentration and relative humidity were placed in the sample introduction system without ozone scrubber as reference samples.





## 2.6 Permeation method

An adapted sample introduction system (Mayer and Borsdorf, 2014) was used to generate air samples containing different
amounts of analytes, levels of relative humidity and ozone concentrations (Figure S2). For each substance, a permeation vessel
was prepared by adding about 500 µL of the substance into a GC vial and closing it with a polyethylene membrane. The
permeation vessel containing Linalool was additionally equipped with a perforated septum to lower the permeation rate. The
four permeation vessels were placed pairwise in two glass tubes which were continuously flowed through with Nitrogen (5.0
grade) at a rate of 500 mL/min. Defined partial flows from both glass tubes were introduced into a mixing chamber via needle
valves and mass flow meters. An additional gas flow of Nitrogen (5.0 grade) was introduced via a mass flow controller for
further dilution of the gas stream. An aliquot of the combined and diluted gas stream was transferred to a second mixing
chamber using a needle valve and a mass flow meter. In this second mixing chamber, the substance-containing gas stream was
further diluted with humidified and ozone-containing air. Parts of the final combined air stream were continuously sucked
through the sorbent tubes for four hours with a constant flow of 80 mL/min using a sample extraction setup containing mass
flow controllers and a vacuum pump.  Temperature, relative humidity and ozone concentrations of the final gas stream were
measured before and after the enrichment experiments (Table S4). The masses of all reference substances enriched per sample
were calculated based on their permeation rates, the dilution setup and the total volume of enriched air, resulting in mean
values (±SD) of $21.58 \pm 1.93$ ng α-Pinene, $27.43 \pm 2.02$ ng Limonene, $29.52 \pm 8.74$ ng Linalool and $24.71 \pm 1.97$ ng Myrcene.
Permeation rates were calculated based on the mass losses of each permeation vessel over a time of at least 24 hours. All used
mass flow controllers and mass flow meters were calibrated using a bubble flow meter before use.

## 2.7 Application of potassium iodide ozone scrubbers under real-world conditions

Two MTS-32 multi-tube sequential samplers (Markes International Ltd., UK) were operated in parallel at the SMEAR II site
at the Hyytiälä Forestry Field Station (Finland) for a period of two consecutive days in September 2024. Forest air was enriched
using Bio-Monitoring sorbent tubes (Markes International Ltd., UK) for 4 hours per sample and a flow of 80 mL min[-1] above
canopy level at 35 m height. In order to evaluate the influence of the ozone scrubber, one sampler was operated with KI-loaded
filters and one without ozone filters. Ozone concentration, temperature, relative humidity and precipitation were monitored
during the sampling period using the SMEAR II site's monitoring infrastructure and extracted from the SmartSMEAR database
(Junninen et al., 2009).

## 2.8 TD-GC-MS analysis

A GC-MS system consisting of an 8890 gas chromatograph and a 5977C single quadrupole mass spectrometer (Agilent
Technologies, Inc., USA) was used for terpene analysis. A TD100-xr thermal desorption system (Markes International Ltd.,
UK) was used to introduce the samples into the instrument. The sample tube was dry purged before desorption for 2 min with
a flow of 20 mL/min to remove water. Tube desorption onto a focussing trap (General Purpose, Markes International Ltd.,



UK) was performed at 300 °C for 5 min and a flow of 100 mL/min. The focussing trap was purged for 1 min with a purge flow

of 20 mL/min at 20 °C and finally desorbed at 300 °C for 5 min with a column flow of 0.75 mL/min and a split flow of
5 mL/min. Chromatographic separation was done using a VF-5ms column (40 m x 0.15 mm ID, 0.15 µm, Agilent
Technologies, Inc., USA) with hydrogen as carrier gas, a column flow of 0.75 mL/min and the following temperature program:
35 °C for 6.5 min, 6 °C/min to 210 °C, 25 °C/min to 280 °C, 280 °C for 2 min. Mass spectrometric detection was done using
an electron impact ionization (EI) source operated at 70 eV and 230 °C. The quadrupole was operated in scan mode for *m/z* 35-

500 at 150 °C. External calibration of the GC-MS system was done using a reference standard mixture. All analytes were
calibrated in a range of 1.5 – 30 ng per sorbent tube. Calibration standards were prepared by injecting a maximum of 20 µL
methanolic standard solutions on Bio-monitoring sorbent tubes according to the procedure described in 2.5. The performance
of the TD-GC-MS system was periodically checked using a weighed reference standard mixture.

**2.9 Data analysis**

GC-MS data were analysed using MassHunter Quantitative Analysis (v. 12.0.893.1, Agilent Technologies, Inc., USA). Further
data analysis, statistical tests and data visualisation were performed in RStudio (v. 2023.12.1+402, RStudio Team, 2020) using
R (v. 4.2.2, R Core Team, 2022) with packages ggplot2 (v 3.5.0, Wickham, 2016), ggpubr (v. 0.6.0, Kassambara, 2023), scales
(v. 1.3.0,Wickham et al., 2023b), ggbreak (v. 0.1.2, Xu et al., 2021), tidyr (v. 1.3.1, Wickham et al., 2024), lme4 (v. 1.1-35.5,
Bates et al., 2015), emmeans (v. 1.10.5, Lenth, 2024) and dplyr (v. 1.1.4, Wickham et al., 2023a). The design of the 3D-

printed components and ozone scrubbers, as well as the generation of rendered images, were conducted using the software
Autodesk Inventor Professional 2004.3 (Autodesk Inc., USA).

**3 Results and Discussion**

**3.1 Characterization of ozone filters**

The mass of scrubber material deposited on the filters after drying was $7.57 \pm 0.82$ mg ($0.046 \pm 0.005$ mmol) for KI and

$9.54 \pm 0.86$ mg ($0.038 \pm 0.003$ mmol) for $Na_2S_2O_3 \cdot 5\,H_2O$ (Table S5). Independent two-sided t-tests were performed to assess
differences between both scrubber types, revealing statistically significant differences in the masses ($p < 0.001$, $n = 20$) and
stoichiometric amounts ($p < 0.01$, $n = 20$) of the scrubber materials deposited on the glass filters. The mean mass of KI
deposited on the glass filters was significantly lower than that of $Na_2S_2O_3$, but the stoichiometric amount of KI was significantly
higher due to its lower molar mass. These results indicate a higher theoretical ozone removal potential of the KI ozone scrubber,

based on the reactions of both materials with ozone (Eq. 1 (Helmig, 1997) and Eq. 2 (Deal et al., 2024; Ernle et al., 2023)).

$$2\,KI + O_3 + H_2O \rightarrow I_2 + O_2 + 2\,K^+ + 2\,OH^- \tag{1}$$

$$2\,S_2O_3^{2-} + O_3 + H^+ \rightarrow S_4O_6^{2-} + O_2 + H_2O \tag{2}$$

To assess the ozone removal potential, the breakthrough of ozone for both filter materials was monitored over a course of 10
hours, which was chosen to reflect realistic sampling durations commonly used in field applications. This duration ensures



that the performance of the filters is evaluated under conditions that closely resemble practical usage while also covering

sufficient time to observe the over-time increase of the downstream ozone concentration. An input ozone concentration of

$50 \pm 3$ ppb was used, with different levels of relative humidity tested to assess their influence on the ozone removal efficiency

(Figure 2).

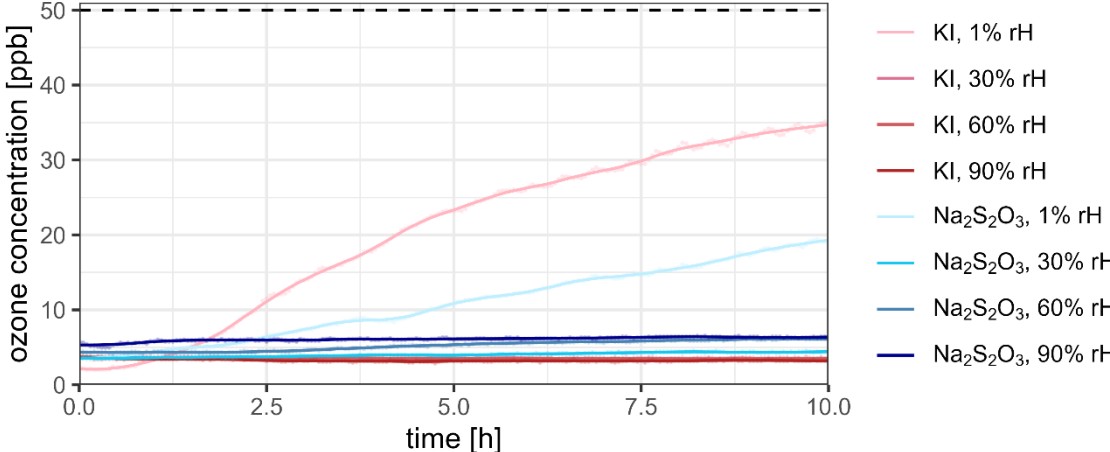

**Figure 2. Progression of the ozone concentration downstream of the ozone scrubbers at different relative humidity levels and an input ozone concentration of $50 \pm 3$ ppb (dashed horizontal line) and a flow rate of 80 mL/min per filter. Potassium iodide (KI, red graphs) and Sodium thiosulfate ($Na_2S_2O_3$, blue graphs) were used as scrubber materials.**

At a very low relative humidity level, both scrubber materials showed an increase in ozone concentration downstream of the

filters in comparison to higher relative humidity levels. This demonstrates that a very low relative humidity inhibits the reaction

of both ozone scrubber materials with ozone, consecutively reducing the ozone removal potential. Relative humidity levels

above 30% result in downstream ozone concentrations of less than than 7 ppb, which correspond to removal efficiencies above

85%. These results are consistent with a previous study, showing that humidity has an influence on the scrubber lifetime of

$Na_2S_2O_3$, with consistently higher ozone removal efficiencies at elevated humidity levels conditions compared to dry air (Ernle

et al., 2023). The highest ozone removal potential was observed for KI with an average of approximately 93% across all

humidity levels. A linear model, formulated as "ozone concentration ~ material * humidity * exposure time", was applied to

the dataset to investigate the influence of scrubber material, relative humidity and time on the downstream ozone concentration.

In order to ensure that the model reflected realistic environmental conditions, the lowest humidity level was omitted. The

analysis demonstrated that both exposure time and humidity significantly affect the downstream ozone concentration, with

differing impacts between the two scrubber materials. For $Na_2S_2O_3$, a 1% increase in relative humidity led to a rise in ozone

concentration by 0.0355 ppb ($p < 0.0001$). In contrast, this effect was significantly smaller for KI (-0.0398 ppb per 1% increase,

$p < 0.0001$), resulting in an almost negligible net effect (-0.043 ppb per 1% rH increase). This indicates that KI scrubbers are

largely unaffected by changes in humidity, while $Na_2S_2O_3$ scrubbers experience a notable decline in performance with

increasing humidity. Similarly, exposure time increased ozone concentration for $Na_2S_2O_3$ by 0.1467 ppb per hour ($p < 0.0001$),

whereas the compared effect was significantly less pronounced for KI (-0.1525 ppb per hour, $p < 0.0001$), resulting in a near-





zero net effect of -0.0058 ppb per hour. This highlights the ability of KI scrubbers to effectively prevent ozone build-up over time. Second-order interactions between humidity and time, as well as the three-way interaction with scrubber material, were not statistically significant, indicating no substantial combined effects on the downstream ozone concentration. The model demonstrated an excellent fit, with a residual standard error of 0.1918 and an adjusted R-squared value of 0.9685, explaining 96.85% of the variance in ozone concentration. The high F-statistic (5294; $p < 2.2e^{-16}$) confirms the model's robustness.

Additionally, a long-term stability test was performed to evaluate the influence of exposure to ambient air on the scrubber's performance. Therefore, the filters were exposed to ambient air with varying ozone concentrations between 6 ppb and 67 ppb and relative humidity levels ranging from 44% to 84 % for 5 and 10 days, representing both typical environmental conditions and timeframe of a sampling campaign (Table S2). After exposure, the breakthrough of ozone was monitored using the same approach as for the freshly prepared filters (Figure 3).

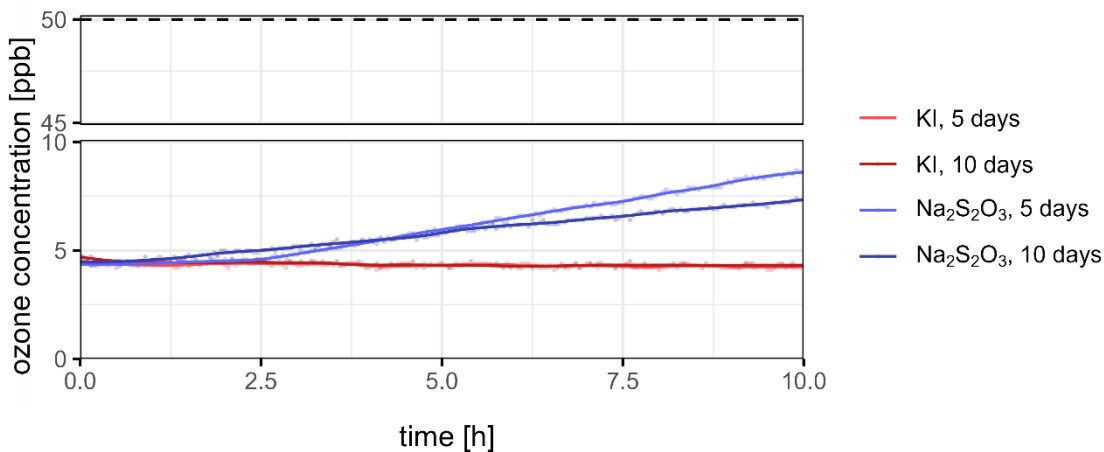


**Figure 3. Progression of the ozone concentration downstream of the ozone scrubbers at 60% relative humidity and an input ozone concentration of 50 ± 3 ppb (dashed horizontal line) and a flow rate of 80 mL/min per filter. Potassium iodide (KI, red graphs) and Sodium thiosulfate ($Na_2S_2O_3$, blue graphs) scrubbers were stored under environmental conditions for 5 and 10 days before the breakthrough tests.**

The measured downstream ozone concentration after 5 and 10 days of exposure to ambient air remained at a consistently low level for the KI-loaded filters, demonstrating an ozone removal efficiency of over 90 % comparable to freshly prepared filters. In contrast, the downstream ozone concentration of the $Na_2S_2O_3$-loaded filters showed an over-time increase after both 5 and 10 days of exposure, leading to a diminished ozone removal efficiency of about 84 %. However, for short sampling times less than 2 hours, the ozone removal efficiency still exceeded 90 %. These results indicate that, in contrast to $Na_2S_2O_3$, the KI-

loaded ozone scrubbers' efficiency is not significantly affected by exposure to ambient air for up to 10 days. This is in accordance with a previous study, showing a higher ozone removal potential of KI compared to $Na_2S_2O_3$ (Fick et al., 2001). However, both scrubber materials show sufficiently high ozone removal efficiencies, as shown in previous studies, especially for short sampling durations (Ernle et al., 2023; Hellén et al., 2024).



Additionally, the influence of the copper wool on the downstream ozone concentration was investigated at an input ozone
concentration of 50 ppb across different humidity levels over the course of 4 hours using cleaned glass filters without any
scrubber material (Figure 4).

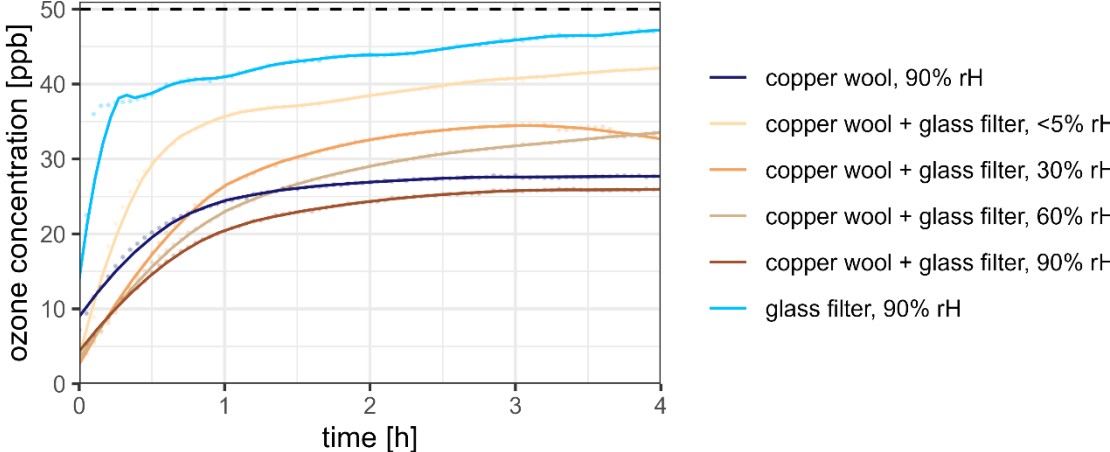

**Figure 4. Progression of the ozone concentration downstream of the copper wool and clean glass filters at different relative humidity levels and an input ozone concentration of 50 ± 3 ppb (dashed horizontal line) and a flow rate of 80 mL/min per filter over the course**
**of 4 hours, representing a typical sampling duration. Only copper wool (dark blue graph) and only clean glass filters (light blue line) were used as reference.**

Our results demonstrate, that the used copper wool has an additional positive effect on the ozone removal potential of the
scrubber. At a relative humidity level of 90%, clean glass filters reduce amount of ozone passing through by approximately
14% over the course of 4 hours. This reduction may be attributed to the adsorption and dissolution of ozone in water films
forming within the porous structure of the glass filter (Willis and Wilson, 2022). Using only copper wool at the same
conditions, the amount of ozone passing through is reduced by approximately 50%, which indicates an ozone-depleting
reaction at the surface of the copper wool. When combining copper wool and clean glass filters, the reduction of downstream
ozone increases with rising relative humidity. At <5% rH, ozone levels decrease by 28%, while at 90% rH, the reduction
reaches 56%. This suggests a humidity-dependent ozone removal mechanism, likely involving the formation of $Cu_2O$ by wet
oxidation of Cu and further reaction between the $Cu/Cu_2O$ system and ozone (Kim et al., 2023; Lin and Frankel, 2013; Ma et
al., 2024). This is supported by the optical changes observed after the breakthrough experiments at high relative humidity,
where the copper surface lost its metallic lustre and developed a faint, dull reddish-brown tint, indicating the formation of
$Cu_2O$. Furthermore, the 56% reduction in downstream ozone observed with the combination of copper wool and a glass filter
suggests a cumulative effect. The glass filter initially removes 14% of the ozone, followed by an additional 50% reduction by
the copper wool, theoretically leading to a total removal of 57% of the initial ozone passing through the system.

In summary, while both scrubber materials demonstrate efficacy, the KI-loaded ozone scrubbers demonstrated a slightly
superior performance in ozone depletion compared to the $Na_2S_2O_3$-loaded glass filters with consistent removal efficiencies
> 90%, characterized by a lower initial downstream ozone concentration, a diminished increase in ozone concentration over





time, a less pronounced influence of varying relative humidity levels during sampling and a reduced propensity to be negatively
affected by exposure to ambient air. These findings indicate that both ozone scrubber materials are generally capable of
removing ozone from the sampled air stream across a range of relative humidity levels. Furthermore, the copper wool
demonstrated an additional ozone-depleting effect, enhancing the overall ozone removal potential of the scrubber design.
Therefore, the proposed scrubber design, where each sorbent tube is equipped with an own ozone filter, is especially suitable
for the long-term environmental monitoring using multi-tube samplers, since the exposure to ambient air of varying
environmental conditions for a longer period of time does not diminish the ozone removal potential when KI is used.
Consequently, there is no need for regular checks and replacement during a monitoring campaign, in contrast to other ozone
filter designs (Hellén et al., 2024; Pollmann et al., 2005).

**3.2 Influence of ozone scrubbers on terpene sampling under controlled laboratory conditions**

To assess the influence of the ozone filter on the sampling of forest VOCs, α-Pinene (bicyclic monoterpene), Myrcene (acyclic
monoterpene), Limonene (monocyclic monoterpene) and Linalool (acyclic monoterpene alcohol) were selected as reference
compounds to cover a broad range of structural and chemical characteristics (Figure S3). The influence of the filters on these
compounds was evaluated using two different approaches: (A) a load-and-flush method, were the influence of humid ozone-
containing air to already adsorbed molecules is examined, and (B) a permeation method, which covers the gas phase reaction
of the analytes with ozone, the interaction of analytes with humid air and the scrubber material and the possible decomposition
of adsorbed substances by ozonation, representing a realistic sampling scenario. Experimental conditions comprised relative
humidity levels of approximately 5%, 30%, 60% and 90% combined with ozone concentrations of <6, 25 and 50 ppb for each
scrubber material, yielding a total of 24 combinations for each approach. For each approach and each combination of ozone
concentration, relative humidity level and scrubber material, three filtered and one unfiltered sample were generated, resulting
in a total of 192 samples. All experiments were conducted using "Bio-monitoring" sorbent tubes, which contain a combination
of Tenax TA and Carbograph 5TD as sorbent material. This combination offers the advantages of capturing a broader range
of VOCs compared to individual sorbents. The polymer-based Tenax TA is particularly effective for medium- to high-
molecular weight VOCs, like mono- and sesquiterpenes, while the carbon-based sorbent Carbograph 5TD efficiently adsorbs
very light and low-molecular weight VOCs, like Isoprene. This combination is therefore especially suitable for the monitoring
of forest environments, where a broad range of volatile organic compounds is present (Borsdorf et al., 2023; Hellén et al.,
2018; Jaakkola et al., 2024).

**3.2.1 Load-and-flush approach**

Using the load-and-flush approach, the influence of the ozone scrubbers on the recovery rates of already adsorbed analytes
was compared across different ozone concentrations and relative humidity levels with and without the use of an ozone scrubber
(Figure 4).






**Figure 5. Mean recovery rates (± standard deviation) of α-Pinene, Myrcene, Limonene and Linalool for the load-and-flush approach at relative humidity levels of (a) <7 %, (b) 30 %, (c) 60 % and (d) 90 %. Grey bars represent unfiltered samples, red bars potassium iodide filters and blue bars sodium thiosulfate filters.**

Without an ozone filter and at low relative humidity values, the recovery of all substances decreased as ozone concentrations
increased, following the order: α-Pinene < Limonene < Myrcene < Linalool, with Linalool showing the highest recovery loss.
This reduction can be attributed to the reaction of adsorbed analytes with ozone passing the sorbent tube, resulting in analyte
loss. The findings align closely with the predicted reactivity of the analytes with ozone based on their chemical structures and
gas-phase reaction constants, which here serve as a reference for analyte decomposition on the sorbent material (Bernard et
al., 2012; IUPAC, 2024, Figure S3). The degree of decomposition of the substances is expected to increase with the number
of double bonds, while terminal double bonds are less affected than internal double bonds, resulting in the order α-Pinene <
Limonene < Linalool < Myrcene (Calogirou et al., 1996). While our results are in accordance with the expectations for α-
Pinene and Limonene, Linalool shows a slightly higher degree of decomposition compared to Myrcene. This discrepancy may
be attributed to the different nature of gas-phase reactions in comparison to the reaction of adsorbed substances. As humidity
increases, the effect of increasing ozone concentrations is mitigated, indicating that a rise in humidity levels prevents ozone-
analyte interactions, even in the absence of an ozone-depleting reagent. At 90 % rH, no reduction in recovery was observed



with increasing ozone concentrations, indicating that the reaction of ozone with adsorbed analyte molecules is effectively inhibited. This demonstrates, that the water content in the sampled air stream plays a positive role in preventing ozone-analyte interactions inside the sorbent tube. This can partly be attributed to the humidity-dependent ozone depletion by the copper wool, but does not fully explain the high recovery rates at 90% relative humidity.

When either of the scrubber materials was used, the recovery rates at the lowest humidity level also decreased with rising ozone concentrations, but to a generally lesser extent, which is in accordance with the observed reduced ozone removal efficiency of both scrubber materials under low relative humidity levels (Figure 2). Comparing both scrubber materials, the loss of analytes was less pronounced for the KI scrubbers, exhibiting higher recovery rates for Myrcene, Limonene and Linalool, as well as comparable recovery rates for α-Pinene. This is in contrast to the breakthrough experiments, which suggest

a higher ozone removal potential of $Na_2S_2O_3$ at low humidity. This may be explained by a lower relative humidity threshold for effectively filtering ozone of KI compared to $Na_2S_2O_3$, as the breakthrough experiments were conducted at approximately 1 % rH and the load-and-flush experiments were performed at around 5-6 % rH. When the relative humidity level exceeded 30 %, no decrease in recovery rates of the four substances were observed with increasing ozone concentrations. When aggregated across all tested relative humidity levels, no significant differences in recovery was observed between unfiltered

samples at baseline ozone concentrations and filtered samples with either ozone scrubber across ozone concentrations up to 50 ppb (Wilcoxon rank sum test, p > 0.01, Figure S4 a), indicating stable recovery values across a wide range of environmental conditions, therefore enhancing the measurement accuracy and fostering the inter-comparability of samples. However, their efficiency in preventing analyte loss is limited at very low relative humidity levels, consistent with the findings from the breakthrough experiments and a previous study conducted on $Na_2S_2O_3$ (Ernle et al., 2023).

**3.2.2 Permeation approach**

Using the permeation method, the influence of the ozone scrubbers on the recovery rates was compared across different ozone concentrations and relative humidity levels both with and without the use of an ozone scrubber. This approach gives additional information about possible interactions between analytes in the gas phase and the scrubber material while passing the ozone filter at various relative humidity levels and simulates a real-world sampling scenario. Overall, the recovery rates of the

investigated substances were below 100 %, except for α-Pinene at <7 % rH and ozone concentrations <25 ppb (Figure S5). This may be explained by systematic losses resulting from possible interactions between the analytes and the internal surface of the permeation device or the gas-phase interaction between analyte molecules and water or ozone before passing the ozone scrubber. Additionally, increasing humidity in the gas stream may saturate the sorbent material with water, blocking the binding sites and hindering adsorption of analytes, ultimately leading to an overall decreased recovery (Ho et al., 2017;

Wilkinson et al., 2020). Furthermore, possible over- or underestimation of the calculated analyte concentrations in the air stream due to inaccuracies in the weighing method of the permeation vessels may lead to higher or lower recovery values. In order to mitigate these uncertain systematic errors, the recovery rates were normalized to the unfiltered samples at baseline ozone concentration for each humidity level (Figure 5).





**Figure 6. Mean recovery rates (± standard deviation) of α-Pinene, Myrcene, Limonene and Linalool for the permeation approach at relative humidity levels of (a) <7 %, (b) 30 %, (c) 60 % and (d) 90 % normalized to the non-filtered baseline ozone sample for each humidity level. Grey bars represent unfiltered samples, red bars potassium iodide filters and blue bars sodium thiosulfate filters.**

When ozone is present in the sampled air stream and no filter is used, the recovery rates of all substances, except for α-Pinene at >30 % rH, exhibit a noticeable decreasing trend with increasing ozone concentrations, aligning with the sequence observed in the load-and-flush approach (α-Pinene > Limonene > Myrcene > Linalool) and culminating in the complete loss of Linalool at 50 ppb $O_3$ across all humidity levels. With increasing relative humidity, the loss in recovery is mitigated, showing a similar trend compared to the load-and-flush approach, but to a generally lesser extent. This discrepancy may be explained by the additional gas-phase interactions between the analyte molecules and ozone, water vapour and the internal surfaces of the permeation device or the glass filters.

At baseline ozone concentrations <6 ppb, no noticeable differences in recovery rates were observed between samples with and without the use of both scrubber materials across all humidity levels. These results indicate that there is no reaction between the analyte molecules and the ozone-depleting reagents, nor any obstruction of the sampled air's flow path caused by interaction between water vapour and the slightly hygroscopic scrubber materials at elevated relative humidity levels.





When ozone is introduced and one of the scrubber materials was used, the recovery rates of all four substances were generally
higher or comparable to the unfiltered samples across all humidity levels. At an ozone concentration of 50 ppb and low relative
humidity, the use of KI scrubbers led to lower recovery rates in comparison to $Na_2S_2O_3$, which is the opposite of the load-and-
flush approach, indicating the limited reliability and efficiency of both scrubber materials at low relative humidity levels <7 %.
When aggregated across all humidity levels ≥30 %, no significant differences in recovery rates were observed between
unfiltered samples at baseline ozone concentrations and filtered samples with either scrubber material and ozone concentrations
up to 50 ppb (Wilcoxon rank sum test, $p > 0.01$, Figure S4 b).

Our results clearly highlight the extent of the negative influence of ozone on the measurement accuracy, emphasizing the
necessity of using a suitable ozone filter. Both KI and $Na_2S_2O_3$ proved to be generally well-suited ozone scrubber materials for
sampling BVOCs in forest air, as they do not exhibit any negative effect on the various terpenes and show no adverse impact
on analyte retention across typical relative humidity levels present in forests. Furthermore, both scrubber materials ensure that
the measurement results accurately reflect the actual terpene concentrations present during sampling, thereby enhancing the
reliability of terpene monitoring in forest air under different and changing environmental conditions.

### 3.2.2 Comparison of both validation approaches

A combination of statistical methods was employed to assess the influence of ozone, relative humidity and the ozone scrubbers
on the recovery rates of α-Pinene, Myrcene, Limonene, and Linalool in both the permeation and the load-and-flush approaches.
A linear model, defined as "Recovery ~ ozone * humidity * filter", was used to determine the direction and magnitude of
individual factors and interaction among these factors. Analysis of Variance (ANOVA) was performed to partition the total
variance and assess the statistical significance of the main factors and interactions. Additionally, Tukey's Honest Significant
Difference test (TukeyHSD) was applied to compare the effects of different scrubber materials (Table 1). Effect sizes were
compared across both methods using the partial Eta-squared (Figure S6). This integrated statistical approach provides a
comprehensive and robust evaluation of the factors affecting the recovery rates across both methods and serves as the
foundation for comparing the two approaches in terms of their suitability for validating ozone scrubbers and assessing their
impact on analyte recoveries.





**Table 1. Results of the statistical analysis evaluating the effects of ozone (O3), relative humidity (rH), ozone scrubbers, and their interactions on the recovery rates of α-Pinene, Myrcene, Limonene and Linalool in both the permeation and load-and-flush approaches. The upper section presents the outcomes of the linear model (Recovery ~ ozone * humidity * filter) and ANOVA, showing the directions of effects (increase ↑, decrease ↓, mixed effect ↕ and near-zero effect –) and their significance (\*\*\* p < 0.001, \*\* p < 0.01, \* p < 0.05, . p < 0.1, and n.s. = not significant). The lower section summarizes the results of the Tukey's Honest Significance Difference (TukeyHSD) test, comparing different scrubber materials.**

| Factor | Permeation | | | | Load-and-Flush | | | |
|---|---|---|---|---|---|---|---|---|
| | *α-Pinene* | *Myrcene* | *Limonene* | *Linalool* | *α-Pinene* | *Myrcene* | *Limonene* | *Linalool* |
| **O₃** | ↓ \*\*\* | ↓ \*\*\* | ↓ \*\*\* | ↓ \*\*\* | ↓ . | ↓ \*\*\* | ↓ \*\*\* | ↓ \*\*\* |
| **rH** | ↓ \* | – \*\*\* | ↓ \*\* | ↓ \*\*\* | ↑ \*\* | ↑ \*\*\* | ↑ \*\*\* | ↑ \*\*\* |
| **scrubber** | ↓ \*\*\* | ↓ \*\*\* | ↓ \*\*\* | ↓ \*\*\* | n.s. | ↓ \*\*\* | ↓ \*\*\* | ↕ \*\*\* |
| **O₃ x rH** | – \*\* | ↑ \*\* | ↑ \*\* | ↑ \*\* | – \*\* | – \*\*\* | – \*\*\* | – \*\* |
| **O₃ x scrubber** | n.s. | ↕ \*\*\* | ↕ \*\*\* | ↕ \*\*\* | n.s. | ↓ \*\* | ↓ \*\* | ↓ \*\* |
| **rH x scrubber** | n.s. | ↕ . | n.s. | n.s. | n.s. | ↑ \*\* | ↑ \*\* | ↕ \*\* |
| **rH x O₃ x scrubber** | n.s. | ↓ \*\* | ↓ \*\* | ↓ \* | n.s. | n.s. | ↑ . | n.s. |
| | comparison of different filters (TukeyHSD) | | | | | | | |
| **Na₂S₂O₃ vs. KI** | ↓ \* | n.s. | n.s. | n.s. | n.s. | n.s. | n.s. | n.s. |
| **unfiltered vs. KI** | ↓ \* | ↓ \*\*\* | ↓ \*\*\* | ↓ \*\*\* | n.s. | ↓ \* | ↓ \* | ↓ \* |
| **unfiltered vs. Na₂S₂O₃** | n.s. | ↓ \*\*\* | ↓ \*\*\* | ↓ \*\*\* | n.s. | ↓ \* | ↓ \* | ↓ . |

In both approaches, an increase in ozone concentration results in a highly significant decrease in the recovery of all analytes, except for α-Pinene in the load-and-flush approach, where the effect is only marginally significant. This highlights the degradation of analytes due to ozonation, both during sampling and after adsorption on the sorbent material.

In contrast, increasing relative humidity leads to contrasting effects in the two methods. In the permeation approach, higher relative humidity leads to a significant decrease in recovery for all substances except Myrcene, which remains largely unaffected. This effect can be attributed to the condensation of water vapour within the sorbent tube, forming a water layer on the sorbent material and consecutively inhibiting the access of analytes to the binding sites (Ho et al., 2017; Wilkinson et al., 2020). Conversely, in the load-and-flush approach, the recovery rates for all substances increase with rising humidity. This effect may also be explained by the formation of a water layer that inhibits the interaction of gas-phase ozone to the already adsorbed analyte molecules, thereby mitigating ozone-induced degradation. Additionally, the effect size of relative humidity and interaction terms including humidity is considerably larger in the load-and-flush approach compared to the permeation method, indicating a stronger influence of humidity-related processes in this setup.

In the permeation method, the filters exhibit stronger and more statistically significant effects on the analyte recovery compared to the load-and-flush approach. This, along with the different effects of relative humidity on analyte recovery, suggests that





the permeation method is better suited to evaluate the efficiency and suitability of the ozone scrubbers, since this method more closely reflects a real-world sampling scenario.

## 3.3 Influence of potassium iodide ozone scrubbers on terpene sampling under real-world conditions

Forest air was monitored for terpenes over a course of 48 hours under changing environmental conditions. The temperature ranged from 12.9 °C to 20.4 °C with relative humidity levels between 47 % and 100 % and ozone concentrations from 21 ppb to 51 ppb. To evaluate the influence of the KI ozone scrubber, the measured terpene concentrations with and without the use of a scrubber were compared (Figure 6 a – c). Additionally, the peak areas of known Tenax degradation products for both scenarios were evaluated (Figure 6 d – f, Klenø et al., 2002).

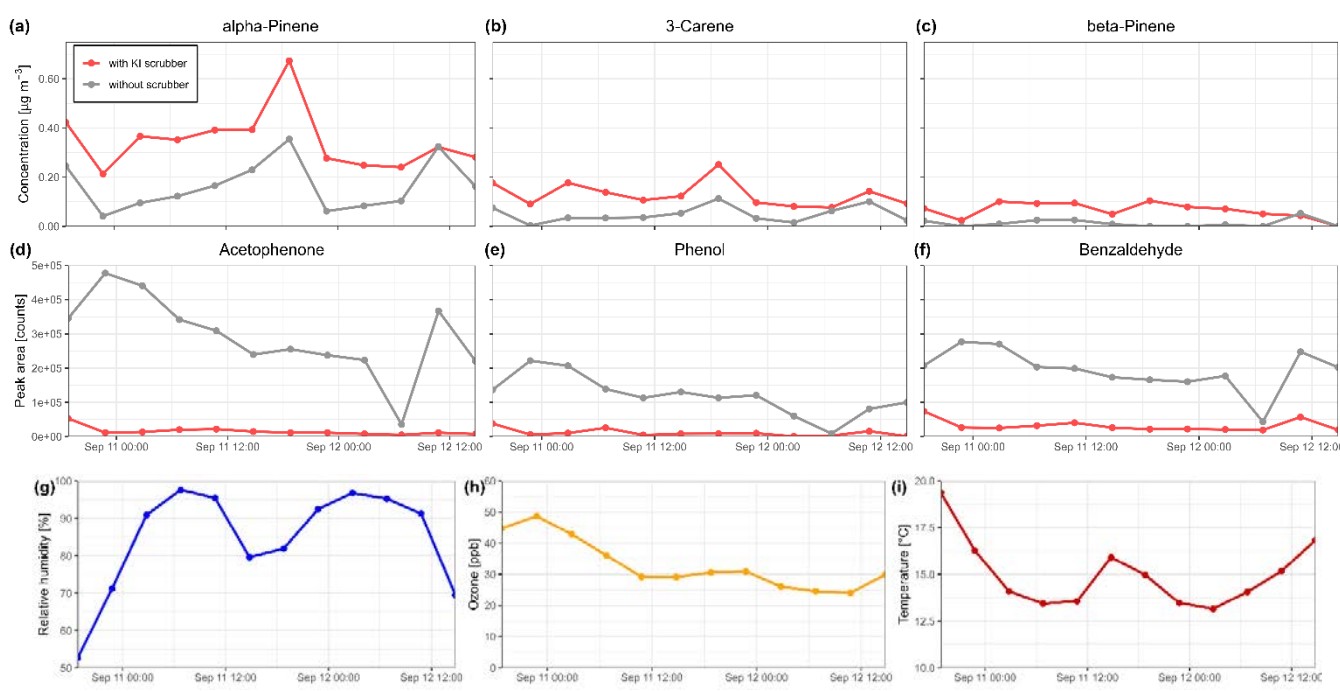

**Figure 7. Temporal variations in monoterpene concentrations (a – c), peak areas of Tenax degradation products (d – f) and environmental parameters (g – i) across the sampling period, using a potassium iodide ozone scrubber (red) and without the scrubber (grey). The results demonstrate that the ozone scrubber enhances the measured concentrations of forest monoterpenes and reduces the formation of degradation products of the sorbent material, indicating effective prevention of reactions of adsorbed monoterpenes and sorbent material with ozone.**

The measured concentrations of monoterpenes (α-Pinene, 3-Carene, β-Pinene) showed significant differences between filtered and unfiltered samples, with higher concentrations in filtered samples (Wilcoxon rank sum test, $p < 0.01$). This demonstrates the positive effect of the used ozone scrubber on the measurement accuracy by preventing analyte loss due to interaction with ozone inside the sorbent tube, consistent with the findings from the laboratory experiments. No significant correlations between terpene concentrations and environmental parameters (humidity, temperature, ozone, precipitation, photosynthetically active radiation) were found. However, a spike in terpene concentrations can be sensed after temperature increased and relative





humidity levels decreased. 3-Carene and α-Pinene showed a significant positive correlation ($\rho = 0.91$, $p < 0.05$), indicating a similar emission pattern and dependency on environmental factors, as shown in a previous study (Borsdorf et al., 2023).

In contrast to the monoterpenes, the detected amounts of the Tenax degradation products Acetophenone, Phenol and
Benzaldehyde were significantly higher in the unfiltered samples (Wilcoxon rank sum test, $p < 0.01$), indicating the effective removal of ozone from forest air under varying conditions and preventing ozone-sorbent interactions. For the unfiltered samples, a Spearman correlation analysis revealed significant positive correlations of the degradation products with ozone, with correlation coefficients of 0.73 for Acetophenone, 0.86 for Phenol and 0.60 for Benzaldehyde ($p < 0.05$), demonstrating the negative influence of ozone on the sorbent material.

These results demonstrate that the KI ozone scrubbers effectively enhance the quantification of terpenes in forest air under varying real environmental conditions, while safeguarding the sorbent material from degradation. Thus, the results of the laboratory experiments could be confirmed under real-world conditions.

**4 Conclusion**

In conclusion, both KI and $Na_2S_2O_3$ showed a sufficient ozone removal performance across environmentally relevant levels of
relative humidity and ozone concentration. A direct comparison between both scrubber materials showed, that KI has a slightly better performance and is less affected by increasing relative humidity levels compared to $Na_2S_2O_3$. Additionally, KI shows a enhanced performance after exposure to ambient air for up to 10 days, with ozone removal efficiencies $> 90\%$, which proves its long-term stability and demonstrates that regular inspections and replacement during longer monitoring campaigns is not necessary. No negative effect of both scrubber materials on four structurally different monoterpenes (α-Pinene, Myrcene,
Limonene, Linalool) was observed across different relative humidity levels and ozone concentrations using a load-and-flush and a permeation approach, demonstrating the suitability for monitoring of terpenes in forest air. At ozone concentrations of 25 ppb and 50 ppb, the use of both ozone scrubber types resulted in recovery rates comparable to measurements without a filter at baseline ozone concentrations and the same relative humidity level. Therefore, the use of the novel ozone filter increases measurement accuracy and improves the comparability of measurements under different environmental conditions.
In a field-test scenario, the KI-loaded scrubbers were demonstrated to enhance the detection of forest monoterpenes under environmental conditions while safeguarding the sorbent material. This results in more accurate measurements and an increased longevity of the sorbent, due to the prevention of reactions of ozone with adsorbed analytes and the sorbent material. These findings, in conjunction with the compatibility of the ozone scrubber design with multitube samplers, the rapid and straightforward replacement combined with the reusability of the glass filters, render this filter design a very well-suited choice
for the environmental monitoring of VOCs without safety concerns and the need of additional devices and power supply. Additional optimization of the filter design may be possible by introducing a drying agent between the ozone filter and the sorbent bed to also mitigate the negative effect of high relative humidity levels on the sorption efficiency (Maceira et al., 2017). Furthermore, the different effects of relative humidity in of both the Load-and-Flush and the permeation approach demonstrate



that the comparatively simple Load-and-Flush approach is not sufficient to assess the suitability of ozone filters for the use in
environmental monitoring.

*Data availability.* Data are available upon request by contacting the corresponding author (robby.rynek@ufz.de).

*Author contributions.* **RR**: Conceptualization, Methodology, Formal analysis, Investigation, Visualization, Writing – Original
draft. **TM**: Conceptualization, Methodology, Resources, Visualization, Writing – Review & Editing. **HB**: Conceptualization,
Methodology, Resources, Funding Acquisition, Writing – Review & Editing.

*Competing interests.* The authors declare that they have no conflict of interest.

*Acknowledgements.* The authors would like to thank Anne Kretzschmar (UFZ) for her assistance in conducting the laboratory
experiments and the preparation of field experiments as well as the team of the UFZ workshop for building the ozone scrubber
cases. Furthermore, we thank the personnel of the Hyytiälä Forestry Field Station, especially Lauri Ahonen and Ilona
Ylivinkka, for their assistance during the field experiment and their effort in the maintenance of the instrumentation.
Additionally, we acknowledge Petri Keronen and Pasi Kolari (both University of Helsinki) for the provision of environmental
data.

*Financial support.* This research has been supported by the German Federal Ministry of Food and Agriculture (BMEL) and
the German Federal Ministry of the Environment, Nature Conservation, Nuclear Safety and Consumer Protection (BMUV)
within the framework of the Forest Climate Fund (grant number 2220WK15B4). Additionally, this work has received financial
support by the H2020 project eLTER PLUS, GA 871128, and acknowledges support from University of Helsinki of the
Hyytiälä SMEAR II LTER site (https://deims.org/663dac80-211d-4c19-a356-04ee0da0f0eb).

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
