# Peer review of "Enhancing forest air sampling using a novel reusable ozone filter design"

_EGUsphere, 2025_

## Author Response (AR1)

**List of revisions in manuscript egusphere-2025-1387**

**Title:** "Enhancing forest air sampling using a novel reusable ozone filter design"

**Authors:** Robby Rynek, Thomas Mayer, and Helko Borsdorf

**Reviewer #1 – Junfeng Liu**

| Reviewer comments | Author response | Revised text as it appears in the manuscript |
|---|---|---|
| *Figure 2 is not clear. Using different color, so in Figure 4.* | In the revised manuscript, we slightly adjusted the color scheme of Figures 2 and 4 to make the graphs more distinguishable. | The color scheme of Figures 2 and 4 was slightly adjusted to increase the distinguishability between the graphs. |
| *Figure 3. The time is so short for the ozone is raising quickly for the Na2S2O3-loaded filter.* | Indeed, the downstream ozone concentration starts to increase during the first two hours of flushing with air (60% rH, 50 ppb O3) at 80 mL/min. This is in contrast to the KI-loaded filters, which showed stable ozone concentrations at baseline level over the course of 10 hours. Nevertheless, the overall increase of the ozone concentration from around 5 ppb to around 9 ppb over the course of 10 hours is still relatively low, given the input concentration of 50 ppb. | No changes made. |
| *Figure 5. The result of the recovery rates of linalool at RH<7% is very difficult for understand, please repeat this experiment to ensure the correctness and repeatability of the experimental results.* | This experiment was already repeated to ensure the correctness. In both cases, a decrease of the Linalool concentration with increasing ozone concentration was observed, both with and without the use of one of the ozone filters. The comparably high variations in the triplicates was also observed, showing that both ozone filters are not reliably depleting ozone under these conditions. | No changes made. |

**Reviewer #2 – Anonymous**

| Reviewer comments | Author response | Revised text as it appears in the manuscript |
|---|---|---|
| *Page 7 Line 191: Probably I haven't read that. How do you know that both samplers perform in the same way?* | Both samplers are identical MTS-32 multi-tube sequential samplers (Markes International Ltd., UK), set up with the same parameters (sampling time, pump flow rate) and were operated in parallel next to each other. Additionally, the flow through all tube positions of both samplers was verified to be equal (±5%) before the experiment using a flow meter (7000 GC flowmeter, Ellutia Ltd., UK). This description was added to the revised manuscript. | Lines 193f now read: "In order to evaluate the influence of the ozone scrubber, one sampler was operated with KI-loaded filters and one without ozone filters. To ensure comparability, both samplers were operated using the same parameters (sampling time of 240 min, flow rate of 80 mL/min) and the flow through all tube positions was verified to be equal (± 5%) using a flow meter (7000 GC flowmeter, Ellutia Ltd., UK)." |
| *Page 12 Line 320 ff: Four different humidities, three ozone concentrations each, and two scrubbers amount to 24 measurements, right. For each set, you measured three filtered and one unfiltered. ...I don't get the 192.* | To evaluate the ozone filters, we used two scrubber materials (KI, Na2S2O3), two approaches (load-and-flush, permeation), each at four relative humidity levels (<7%, 30%, 60%, 90%) with three ozone concentrations (baseline, 25 ppb, 30 ppb). For each experiment, three samples and one control sample (without filter) were generated, resulting in a total number of 192 samples. | No changes made. |
| *Figure 3: Add in the caption "…over the course of 10 hours."* | Thank you for pointing out the missing information. The figure capture was modified according to your suggestion. | Figure caption 3 now reads: "Progression of the ozone concentration downstream of the ozone scrubbers at 60% relative humidity and an input ozone concentration of 50 ± 3 ppb (dashed horizontal line) and a flow rate of 80 mL/min per filter over the course of 10 hours. (…)" |
| *Results in Figures 5 and 6: Do you have any idea why the recovery rate is that much above 100%?* | Recovery rates exceeding 100% in the load-and-flush approach are likely attributable to inaccuracies in the preparation of the | No changes made. |

| | custom-made stock solution used for the experiments and the application of the diluted standard solution using a microliter syringe. In contrast, the calibration curves used for quantification were based on a commercially available, certified stock solution, offering greater accuracy. | |
| | In the evaluation of the permeation experiments, recovery rates were normalized to those of the unfiltered samples at baseline ozone concentrations at the corresponding relative humidity levels. As a result, values exceeding 100% are artifacts of the normalization procedure. Without normalization, the recovery rates were below 100%, except for α Pinene at <7% rH and <25 ppb O3, as detailed in lines 382f and illustrated Figure S5. | |

**Reviewer #3 – Anonymous**

| Reviewer comments | Author response | Revised text as it appears in the manuscript |
|---|---|---|
| *Line 119 / Figure Capture 1: The layout scheme of the ozone scrubber needs to be equipped with markers (a, b, c, d...) of the parts that make it up, such as the scrubber housing, glass filter, copper wool, two O-rings, etc., and can also be marked with the direction of the air sample entering the ozone scrubber.* | We adjusted the schematic layout of the ozone scrubber design according to your suggestion and added markers for the individual parts. The figure caption was adjusted accordingly. | Markers and Arrows were added to Figure 1. Figure caption 1 now reads: "Schematic layout of the ozone scrubber. The scrubber housing (A) was designed to fit on a commonly used diffusion-locking cap (B). Sample air flows from the left to the right, passing both a loaded glass filter as ozone scrubber (C) and copper wool (D). A screw cap (E) facilitates the quick and easy exchange of the filter. Two O-rings (F), located between the screw cap and the filter and between the housing and the diffusion-locking cap, prevent sampled air from bypassing the filter. Arrows indicate the direction of the air flow through the ozone filter." |
| *Line 124f: In the preparation of ozone scrubber, the method of loading glass filters with fresh KI or Na2S2O3 solution has not been explained in detail, is the glass filter only rinsed with the solution until it is completely wet (how to ensure that it is completely wet) or is the glass filter soaked for some time? The method of making fresh KI and Na2S2O3 solutions needs to be explained here?* | The glass filters were placed in a filter holder and rinsed with the respective salt solution until complete wetting, as described in lines 126f. We expanded the experimental description for loading the ozone filters and added the preparation of the salt solutions. | Lines 125f now read: "Ozone scrubbers were prepared by loading cleaned and dried glass filters (VitraPOR, diameter 9 mm, height 3.5 mm, pore diameter 100 - 160 μm, ROBU Glasfilter-Geräte GmbH, Germany) with freshly prepared solutions of KI or $Na_2S_2O_3$. The solutions were prepared by weighing 40 g of the respective salt and subsequent dissolution with 360 g of Milli-Q water, resulting in a concentration of 10% w/w. For loading the ozone filters, the glass filters were (…)" |
| *Line 142: How the flow rate requirements for ozone monitors need to be detailed here?* | The flow requirements of the ozone monitor (>600 mL/min) were added in the revised manuscript. | Line 145f now reads: "(…) 3D-printed filter holder (Figure S1), which was designed to |

| | | meet the flow requirements (>600 mL/min) of the ozone monitor" |
|---|---|---|
| *Line 142f: What is the desired flow rate to flow through this glass filter?* | The desired flow rate through the single glass filters (80 mL/min) was added in the revised manuscript. | Line 145 now reads: "to ensure a homogeneous flow through all filters (80 mL/min each)." |
| *Line 165 / Tables S3 and S4: In the experimental column in Tables S3 and S4, several/variations of experimental conditions were performed, but what is meant by 0-0, 0-25, 0-50, 30-0, 30-25, 30-50, (what does the first number indicate - what does the second number indicate?)* | An explanation of the experimental conditions was added to the "Experiment" columns in Tables S3 and S4. Additionally, the experiments at low relative humidity were renamed to "<7 – 0", "<7 – 25" and "<7 – 50" to be consistent with the conditions named throughout the manuscript. | Tables S3 and S4 now read:

**Experiment (rH – O$_3$)**
<7 – 0
<7 – 25
<7 – 50 |
| *Line 217 / Results and Discussion: In the Characterization of ozone filters subsection in material methods, it is explained about terpene analysis using TD GC-MS. However, in the result and discussion section, it has not been explained how the results of terpene analysis using GC-MS (typical separation chromatogram produced from GC-total ion chromatogram, typical mass spectrum of each terpenes analytes), so that it can confirm good component separation and proper component identification.* | For the purpose of the manuscript, a detailed description of the TD-GC-MS results, aside from the measured concentrations, is not the objective and would be beyond the scope of this paper. We therefore prefer to not include this detailed information in the manuscript. Nevertheless, the used TD-GC-MS system was calibrated using a terpenes standard solution (Terpenes MegaMix Standard #1, Restek GmbH, Germany), showing a sufficient chromatographic separation. The consistency of retention times and mass spectra for the mixture of α Pinene, Myrcene, Limonene and Linalool used in the experiments was also verified prior to conducting the experiments. | No changes made. |
| *Line 239f: Why is it that in low humidity (dry) conditions, the reaction of ozone scrubber material with ozone is inhibited? and vice versa, this phenomenon can be further explained* | Looking at the reactions of the ozone depleting reagents with ozone (Eq. 1 and 2), both reactions require water to take place. While water is a reactant in the ozone depletion reaction using KI, the reaction of Na2S2O3 takes place in an aqueous medium. | Lines 244f now read: "This demonstrates that a very low relative humidity inhibits the reaction of both ozone scrubber materials with ozone, consecutively reducing the ozone removal potential. This diminished ozone removal potential can be explained by the |

| | | |
|---|---|---|
| | When water is present in the sampled air stream, both reactions can occur, with higher efficiency at increasing relative humidity levels. At very low relative humidity levels, both reactions are inhibited due to the lack of a reactant for KI and a suitable reaction medium for Na2S2O3. An explanation of the results was added to the revised manuscript. | role of water in both scrubber – ozone – reactions. While water is a reactant in the reaction of ozone and KI (Eq. 1), the depletion of ozone with $Na_2S_2O_3$ likely takes place in an aqueous medium (Deal et al., 2024). Therefore, the presence of water is necessary to facilitate both ozone depletion reactions." |
| *Line 274: Is there a possibility that ozone scrubber material (KI loaded) still has the same efficiency after 10 days of exposure to ambient air? How is the efficiency of ozone scrubber material after a period of more than 10 days? So that it can be reused in the next time frame sampling campaign* | Yes, the KI-loaded ozone filters still have the same ozone removal potential after 10 days of exposure to ambient air, as shown in Figure 4. Please note that exposure does not mean they are actively flushed with sampled air. Using the same ozone filter twice was not tested, but could be possible based on the breakthrough experiments (10 hours of flushing). Nevertheless, based on our experience from field experiments, we would advise against reusing the filters for a second sampling without applying new reagent, since visual changes (i.e. yellowing) of the filters are observed regularly. | No changes made. |
| *Line 294: The mechanism of ozone removal due to the reaction that occurs with copper wool needs to be explained by adding the reaction mechanism that occurs.* | We added a simplified reaction of copper with ozone in the revised manuscript. The detailed reaction mechanism would go beyond the scope of our work. More detailed information can be found in the cited publications by Kim et al. (2023), Lin and Frankel (2013), and Ma et al. (2024). | Line 303 now reads: "(…) oxidation of Cu and further reaction between the $Cu/Cu_2O$ system and ozone (Eq. 3, Kim et al., 2023; …)

Eq. 3 was added in line 305:
$2\,Cu^0 + O_3 \rightarrow Cu_2O + O_2$ (3) |

**Additional changes made:**

- Eq. 2 was corrected to: $2\,S_2O_3^{2-} + O_3 + 2\,H^+ \rightarrow S_4O_6^{2-} + O_2 + H_2O$
- Numbers of Figures were corrected throughout the manuscript.
- Line 251: "than" was removed.
- Figure S1: The black background was removed.